# Accuracy of artificial intelligence CT quantification in predicting COVID-19 subjects' prognosis

Arvin Arian[1][☯], Mohammad-Mehdi Mehrabi Nejad[1][☯], Mostafa Zoorpaikar[1], Navid Hasanzadeh[2], Saman Sotoudeh-Paima[2], Shahriar Kolahi[1], Masoumeh Gity[1], Hamid Soltanian-Zadeh[2]*

**1** Department of Radiology, School of Medicine, Advanced Diagnostic and Interventional Radiology Research Center (ADIR), Imam Khomeini Hospital, Tehran University of Medical Sciences, Tehran, Iran, **2** Control and Intelligent Processing Center of Excellence (CIPCE), School of Electrical and Computer Engineering, College of Engineering, University of Tehran, Tehran, Iran

☯ These authors contributed equally to this work.
* hszadeh@ut.ac.ir

**Data Availability Statement:** ****PA AT ACCEPT: Please confirm the public availability of Mendeley data at Accept***** All relevant data related to this study are publicly available from the Mendeley

## Abstract

### Background

Artificial intelligence (AI)-aided analysis of chest CT expedites the quantification of abnormalities and may facilitate the diagnosis and assessment of the prognosis of subjects with COVID-19.

### Objectives

This study investigates the performance of an AI-aided quantification model in predicting the clinical outcomes of hospitalized subjects with COVID-19 and compares it with radiologists' performance.

### Subjects and methods

A total of 90 subjects with COVID-19 (men, n = 59 [65.6%]; age, 52.9±16.7 years) were recruited in this cross-sectional study. Quantification of the total and compromised lung parenchyma was performed by two expert radiologists using a volumetric image analysis software and compared against an AI-assisted package consisting of a modified U-Net model for segmenting COVID-19 lesions and an off-the-shelf U-Net model augmented with COVID-19 data for segmenting lung volume. The fraction of compromised lung parenchyma (%CL) was calculated. Based on clinical results, the subjects were divided into two categories: critical (n = 45) and noncritical (n = 45). All admission data were compared between the two groups.

### Results

There was an excellent agreement between the radiologist-obtained and AI-assisted measurements (intraclass correlation coefficient = 0.88, *P* < 0.001). Both the AI-assisted and

Database (http://doi.org/10.17632/pfmgfpwnmm. 1).

**Funding:** The authors received no specific funding for this work.

**Competing interests:** The authors have declared that no competing interests exist.

radiologist-obtained %CLs were significantly higher in the critical subjects (P = 0.009 and 0.02, respectively) than in the noncritical subjects. In the multivariate logistic regression analysis to distinguish the critical subjects, an AI-assisted %CL $\geq$35% (odds ratio [OR] = 17.0), oxygen saturation level of <88% (OR = 33.6), immunocompromised condition (OR = 8.1), and other comorbidities (OR = 15.2) independently remained as significant variables in the models. Our proposed model obtained an accuracy of 83.9%, a sensitivity of 79.1%, and a specificity of 88.6% in predicting critical outcomes.

## Conclusions

AI-assisted measurements are similar to quantitative radiologist-obtained measurements in determining lung involvement in COVID-19 subjects.

## Introduction

Having more than three years elapsed since the first case of COVID-19 was reported, scientists have comprehensively studied the clinical presentations, diagnostic methods, prognostic factors, and treatment options for this novel infectious disease. Owing to its highly contagious nature and intensive care requirements in critical subjects, allocating ICU beds for high-risk subjects is highly recommended to minimize the total adverse events [1]. Clinical presentations, laboratory and imaging findings, and comorbidities were used to predict subjects' clinical progression or outcome [2].

The lungs are predominantly involved in this infectious disease, and the extent of pulmonary involvement has been confirmed to be associated with unfavorable outcomes [3]. To this end, various semi-quantitative and quantitative scoring systems have been developed. These include the chest computed tomography severity score, which identifies patients in need of hospital admission, and the total severity score, which explores the relationship between imaging manifestations and the clinical classification of COVID-19 [4]. These systems, with varying performances, have been used to predict clinical outcomes [5–7]. However, these scoring tools require an expert radiologist to visually investigate all chest CT slices, which increases the analysis time and human errors.

Artificial intelligence (AI) is believed to cause a paradigm shift in healthcare and can be a useful method tussling with the COVID-19 pandemic [8]. Computer-aided quantification of chest CT scans can significantly enhance the sensitivity of measurements in a much shorter time, particularly in countries facing a shortage of radiologists or with radiologists overburdened [9]. It is expected that AI and deep learning technology will significantly improve the management of subjects with COVID-19, especially in diagnosis and prognosis prediction [10].

Taking all of this into consideration, we aimed at investigating the performance of an AI-aided quantification model in comparison to that of a radiologist-obtained measurement method in predicting the clinical outcomes. Further, the best predictive model in combination with clinical and para-clinical findings was proposed.

## Materials and methods

### Study design and population

This cross-sectional retrospective study included a total of 90 subjects (men, n = 59 [65.6%]; age, 52.9±16.7 years; critical, n = 45) scanned using a 16-slice MDCT scanner (Siemens

SOMATOM Emotion, Erlangen, Germany) at Imam Khomeini Hospital, Tehran, Iran. Participants with a positive rRT-PCR finding indicating the presence of COVID-19 underwent chest CT examination between November 2020 and January 2021. Subjects with an uncertain outcome or incomplete required medical data were excluded. All subjects were managed according to the latest national protocol for COVID-19. In addition, deidentification was performed by the data collection team to protect the privacy and confidentiality of the collected subject data. This study was conducted from March 2021 to September 2021 and was approved by the institutional review board and the local ethics committee (IR.TUMS.IKHC.REC.1399.255). The need for written informed consent was waived due to the retrospective design of this study.

CT acquisitions were made in the supine position, at full inspiration and without contrast injection, using a tube voltage of 130 kVp and a tube current-time product of 70 mAs. The scanner had a tube rotation time of 0.6 seconds, and a beam collimation of 1.2 mm. The projection data was reconstructed using Siemens reconstruction toolbox with a mediastinum B20f smooth kernel and a lung B70f sharp kernel with a slice thickness of 5 mm. A slice thickness of 1.2 mm was used to perform sagittal and coronal multiplanar reconstructions.

In order to train the model, we used a dataset that was introduced in a previous study. This dataset was approved under the ethical approval code IR.TUMS.VCR.REC.1399.488, titled "Clinical Feasibility Study of National Teleradiology System for COVID-19" [11]. It consists of 297 subjects (men, n = 167 [56.6%]; age 54.3±19.2 years) with 148 in critical condition.

The subjects were divided into two groups according to their clinical outcomes: (a) critical: subjects who required ICU admission or mechanical ventilation or who expired; and (b) non-critical. For simplicity and clarification, in the rest of the paper, we call the first and second datasets, dataset E and dataset T, respectively.

The data used in this work is publicly available on https://data.mendeley.com/datasets/pfmgfpwnmm [12].

## Data collection

All the following data were retrieved for all subjects: (a) demographic information: sex and age; (b) vital signs: oxygen saturation (SpO2) level, respiratory rate (per minute), blood pressure (BP, mmHg), pulse rate (per minute), and temperature (˚C); (c) immunocompromised conditions: acquired immunodeficiency or hereditary diseases, chemoradiation therapy, or long-term corticosteroid usage; (d) other comorbidities: hypertension, diabetes, pulmonary diseases, or cardiovascular diseases; (e) laboratory findings: white blood cell count, including lymphocyte counts and neutrophil, and hemoglobin, creatinine, platelet, D-dimer, C-reactive protein, vitamin D, procalcitonin, ferritin, and pH levels; and (f) radiological and AI findings, discussed further in the following sections.

## Radiologist-obtained quantification

Two fellowship-trained chest radiologists—with more than 10 years of experience—blinded to clinical data (except for the rRT-PCR results) independently evaluated all CT scans. COVID-19 lesions were manually marked as regions of interest (ROIs) in every cross-section of the lung CT scan using the MRIcro software (https://people.cas.sc.edu/rorden/mricro/mricro.html). The fraction of compromised lung parenchyma (%CL) was calculated as 100 multiplied by the compromised lung parenchyma volume divided by the total lung volume. The intraclass correlation coefficient (ICC) was evaluated for the measurements obtained by the two radiologists to assess the inter-rater reliability.

## AI-assisted quantification

A schematic representation of the workflow is shown in Figs 1 and 2. There were four primary components to our approach: 1) preprocessing of the CT slices, 2) automated segmentation of the infection region on each CT slice using a VGG16-based U-Net model, 3) automated segmentation of each lung using a U-Net model [13, 14], and 4) calculating the fraction of compromised lung parenchyma (%CL).

The GitHub repository of this work is publicly available on: https://github.com/SamanSotoudeh/COVID19-segmentation.

**Image datasets.** Two different datasets have been used in this work. To prevent bias, we have trained and validated our AI-assisted model for segmenting COVID-19 lesions using axial CT slices of 297 subjects from dataset T. Afterwards, the introduced dataset E was used for prognosis analysis and testing of the model.

**Preprocessing.** Preprocessing is a crucial and standard step in medical image segmentation, which accounts for reducing the variability in images. In this study, the image intensities

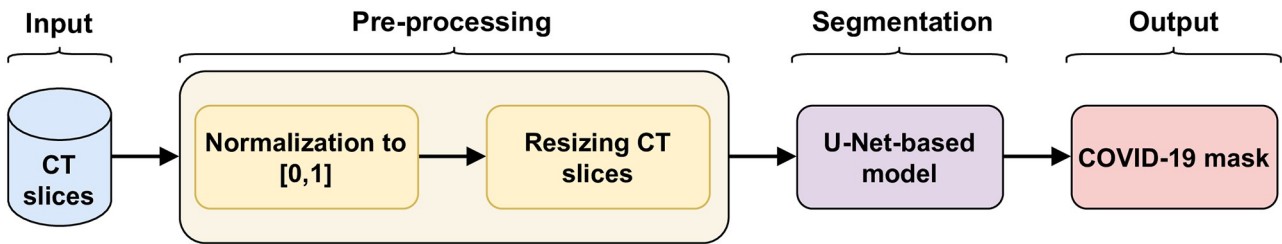

**Fig 1. The COVID-19 segmentation algorithm block diagram.**

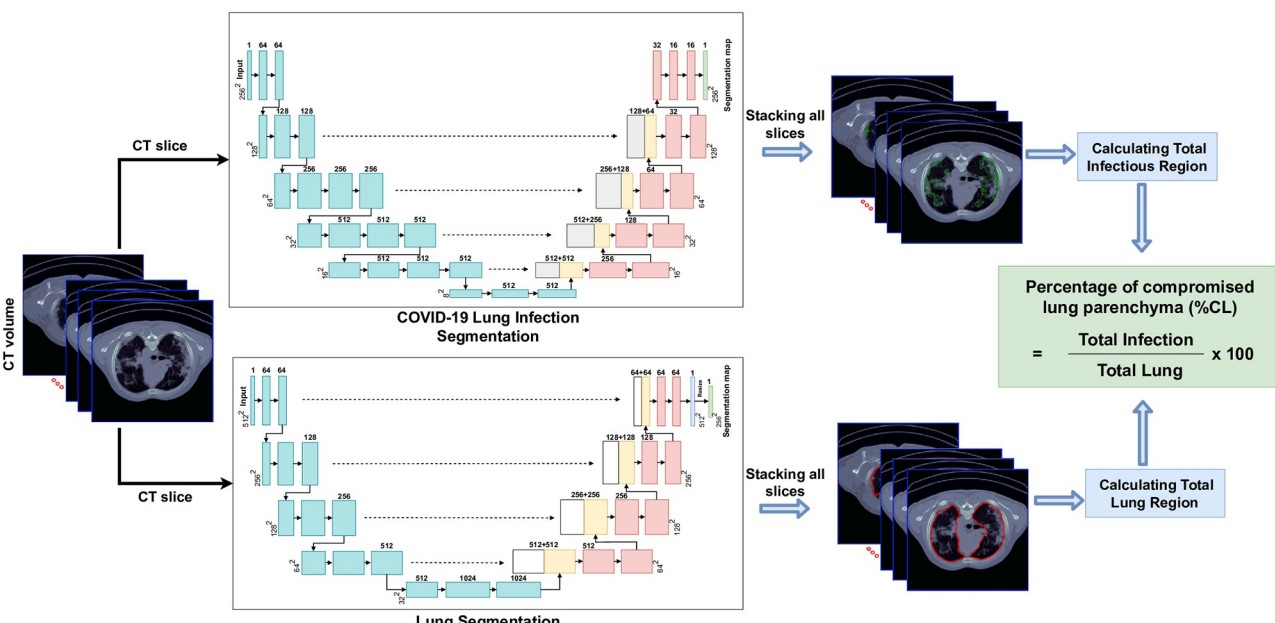

**Fig 2. U-Net architecture and resulting segmentation of the total and compromised lung regions in the CT images and calculation of the %CL (% CL = fraction of compromised lung parenchyma).**

for each slice were normalized in the range of 0 to 1. All the slices were resized to $256 \times 256$ to reduce the computational load and make them suitable for the training procedure.

**Automated COVID-19 infection segmentation.** A deep learning approach based on the U-Net framework (12) was implemented to segment the COVID-19 infection regions on the CT slices. Previous studies have shown that VGG16-based UNet model was successful in COVID-19 lesion segmentation [11, 15–17]. This model can localize abnormal areas in the image and distinguish their boundaries [18]. Moreover, UNet-based models can achieve high accuracy by training on a small dataset with only hundreds of images [14, 19]. Fig 2 shows the architecture of the VGG16-based UNet model used in this work.

**Automated lung segmentation.** For the lung segmentation section, we used an off-the-shelf U-Net model trained on a large and diverse dataset (R231CovidWeb, https://github.com/JoHof/lungmask) [13], which can identify the lung boundary in a couple of seconds. The standard U-Net model's general architecture is similar to the structure described above, except for the block numbers and convolutional layers within each block, which are 4 and 2, respectively.

**Calculating the fraction of compromised lung parenchyma (%CL).** The cumulative lung and COVID-19 infection regions were separately calculated for each lung. We then calculated the fraction by dividing the compromised lung parenchyma volume (Section B.2) by the total lung volume (Section B.3) and multiplying the results by 100.

## Implementation details

Cross-validation is a model validation technique used for evaluating the generalizability of a method on an independent dataset. In this work, a subject-level five-fold cross-validation over the dataset T was conducted for quantitative evaluation. For this purpose, all subjects were randomly split into five parts. Then, at each time, four folds were used as the training set, and one-fold was used for validation. To reduce the expert's segmentation error, we excluded slices of COVID-19 subjects with no observable infection (i.e., only slices with observable infection within the COVID-19 dataset were kept). This slice exclusion strategy eliminated the possibility of missing tiny infection regions. The model still learns the slices with no infection belonging to healthy subjects.

After performing five-fold cross-validation, the best validated model in terms of dice similarity coefficient (DSC) on dataset T was used for prognosis analysis for this study. One of the main limitations of using AI-based models is their limited generalizability. Adopting the approach of using dataset T for training and dataset E for prognosis analysis would provide an unbiased and realistic estimate of the true performance of the AI-based model.

The model was trained using an Adam optimizer. The batch size was 8, and the initial learning rate was $1e - 4$. The learning rate decays by a factor of 0.1 after every three epochs, wherein the validation loss plateaus get to a value of $1e - 7$ at a minimum. An early stopping strategy was used to prevent overfitting if the validation loss did not improve after ten epochs.

The experiment was conducted on a personal computer with Nvidia GeForce GTX 2070 SUPER, with Intel Core i9-7900X CPU.

## Statistical analysis

The IBM SPSS software (version 16, Chicago, IL, USA) was used for performing all the statistical analyses. Qualitative and quantitative variables were reported as frequencies (percentages) and means (standard deviations [SDs]), respectively. The Kolmogorov–Smirnov two-sample test was used to evaluate the normality of data. Association analyses were performed using either the t-test (for continuous variables with normal distribution), Mann–Whitney U test (for continuous not-normal and ordinal variables), or chi-square test (for nominal variables).

P-values of <0.05 were considered statistically significant. ICC—two-way mixed, single measures, absolute agreement—was used to evaluate the inter-rater reliability of the measurements obtained by the two radiologists as well as the AI-assisted %CL and radiologist-obtained %CL.

Variables with P-values <0.1 in the univariate analyses were then included in a backward multiple logistic regression model to adjust for confounding variables. Backward stepwise regression initially introduces all the predictors and then, different predictors are withdrawn one by one till the overall prediction accuracy does not decrease. The odds ratio (OR) for categorical variables is interpreted as the chance of progressing to critical disease when the condition is met. The best predictive model was decided in the final step. Receiver operating characteristic (ROC) curve and Youden's J index [20] was used to define the optimum cut-off values for outcome prediction. The efficiency of the ROC analysis was indicated using the area under the ROC curve (AUC) value. Other measured metrics include accuracy, sensitivity, specificity, positive predictive value (PPV) and negative predictive value (NPV).

## Results

### Study population characteristics

After evaluating 108 subjects, 18 (16.7%) subjects with an uncertain outcome or incomplete required medical data were excluded. A total of 90 subjects met the inclusion criteria, including 59 (65.6%) male subjects with a mean ± SD age of 52.9±16.7 years. Of those, 13 (14.4%) and 43 (47.8%) subjects had a medical history of an immunocompromised condition or other comorbidities, respectively. According to the clinical outcomes, the critical and noncritical groups consisted of 45 subjects each (Table 1).

### Non-radiological findings

The demographic data—age and sex—had no significant association with critical outcomes (*P* = 0.17 and 0.12, respectively). The critical subjects had a significantly lower SpO2 level (*P* < 0.001) and diastolic BP at admission (*P* = 0.003) but a higher temperature (*P* = 0.003) than the noncritical subjects. The subjects with an immunocompromised condition (*P* = 0.007) and those with other comorbidities (*P* = 0.02) were more likely to experience a critical condition. The critical subjects also had significantly higher white blood cell and neutrophil counts but a lower hemoglobin level than the noncritical subjects (*P* = 0.04, 0.01, and 0.001, respectively) (Table 1).

### Image findings

There was an excellent agreement between two radiologists as well as radiologist-obtained and AI-assisted measurements (ICCs = 0.92 and 0.88, both *P* < 0.001). Both the AI-assisted and radiologist-obtained %CLs were significantly higher in the critical subjects (*P* = 0.009 and 0.02, respectively) than in the noncritical subjects. Interestingly, the AI-assisted %CL (AUC = 0.644 [0.53–0.76]; *P* = 0.02) showed a similar AUC to the radiologist-obtained %CL (AUC = 0.639 [0.52–0.75]; *P* = 0.02) (Fig 3).

### Predictive model

Backward multivariate logistic regression was exploited on critical diseases as the outcome of interest and all parameters with a P-value of <0.1 as the independent variables. An AI-assisted %CL of ≥35% (OR = 17.0), SpO2 level of <88% (OR = 33.6), immunocompromised condition (OR = 8.1), and other comorbidities (OR = 15.2) independently remained as significant variables in the models (Table 2). Cut-off values of 35% for the AI-assisted %CL (Youden's J

**Table 1. Demographic, clinical, and paraclinical data of the subjects and the differences between the two groups.**

| Variables | | | | All subjects | Non-critical | Critical | P_value |
|---|---|---|---|---|---|---|---|
| | | | | N = 90 | N = 45 | N = 45 | |
| Demographic Data | Age* | | | 52.9(16.7) | 50.4(17.3) | 55.2(16.0) | 0.17 |
| | Sex | Male | | 59(65.6) | 26(57.8) | 33(73.3) | 0.12 |
| | | Female | | 31(34.4) | 19(42.2) | 12(26.7) | |
| Clinical data | Vital signs * | SpO2 | | 86.0(10.2) | 90.4(6.2) | 81.5(11.6) | <0.001 |
| | | RR | | 24.1(5.5) | 23.2(5.4) | 25.0(5.5) | 0.14 |
| | | Systolic BP | | 128.6(19.1) | 130.6(20.4) | 126.7(17.8) | 0.34 |
| | | Diastolic BP | | 79.9(12.2) | 83.6(13.0) | 76.1(10.1) | 0.003 |
| | | PR | | 95.0(16.4) | 97.2(13.8) | 92.8(18.6) | 0.21 |
| | | Temperature | | 37.7(0.9) | 37.4(0.8) | 38.0(1.0) | 0.003 |
| | Medical history | Immuno-compromised | | 13(14.4) | 2(4.4) | 11(24.4) | 0.007 |
| | | Other comorbidities | All | 43(47.8) | 16(35.6) | 27(60.0) | 0.02 |
| | | | Diabetes | 28(31.1) | 12(26.7) | 16(35.6) | 0.36 |
| | | | Hypertension | 36(40.0) | 17(37.8) | 19(42.2) | 0.67 |
| | | | Chronic heart failure | 6(6.7) | 1(2.2) | 5(11.1) | 0.09 |
| | | | Coronary artery disease | 16(17.8) | 5(11.1) | 11(24.4) | 0.1 |
| | | | COPD | 8(8.9) | 3(6.7) | 5(11.1) | 0.46 |
| | | | Chronic kidney disease | 10(11.1) | 5(11.1) | 5(11.1) | >0.99 |
| Paraclinical data | Laboratory findings * | WBC | All | 8.1(4.3) | 7.1(3.7) | 9.0(4.7) | 0.04 |
| | | | Neutrophil | 6.6(4.0) | 5.6(3.7) | 7.7(4.1) | 0.01 |
| | | | Lymphocyte | 1.1(0.7) | 1.2(0.6) | 1.0(0.7) | 0.24 |
| | | Hemoglobin | | 12.6(2.5) | 13.5(2.2) | 11.7(2.5) | 0.001 |
| | | Platelet | | 225.5(108.9) | 211.5(84.3) | 239.6(128.3) | 0.23 |
| | | Cr | | 1.7(2.0) | 2.0(2.6) | 1.4(0.8) | 0.15 |
| | | D-dimer | | 5714(18658) | 1250(1019) | 9582(25218) | 0.24 |
| | | CRP | | 84.5(68.2) | 79.2(66.2) | 90.7(70.9) | 0.46 |
| | | Vitamin D | | 22.2(8.5) | 26.3(10.8) | 19.6(6.2) | 0.18 |
| | | Procalcitonin | | 3.0(4.3) | 3.1(4.7) | 2.8(4.4) | 0.9 |
| | | Ferritin | | 668.8(766.6) | 600.2(407.0) | 703.2(914.7) | 0.81 |
| | | pH | | 7.4(0.7) | 7.4(0.5) | 7.4(0.8) | 0.08 |
| | Radiologic findings (%CL) | Radiologist measured | | 28.5(19.4) | 23.7(17.7) | 33.3(20.0) | 0.02 |
| | | AI-assisted | | 23.2(17.7) | 18.3(14.4) | 28.0(19.4) | 0.009 |

* Reported as means (standard deviations); all other variables were reported as n (%).

SpO2 = oxygen saturation; RR = respiratory rate; BP = blood pressure; PR = pulse rate; WBC = white blood cell; Cr = creatinine; CRP = C-reactive protein; %

CL = fraction of compromised lung parenchyma; AI = artificial intelligence

index = 0.27) and 88% for the SpO2 level (AUC = 0.81 [0.71–0.90], Youden's J index = 0.61, $P < 0.001$) were defined based on the ROC curve and Youden's J index.

We proposed three models consisting of clinical variables and in combination with radiologist or AI measurements. The combination of clinical and AI findings showed the highest predictive values with a sensitivity of 79.1%, a specificity of 88.6%, a PPV of 87.2%, an NPV of 81.2%, and an accuracy of 83.9% (Table 3).

## Discussion

There is a growing interest in the application of AI in COVID-19 management in the medical community, especially in chest CT analysis. The first efforts were taken to detect COVID-19

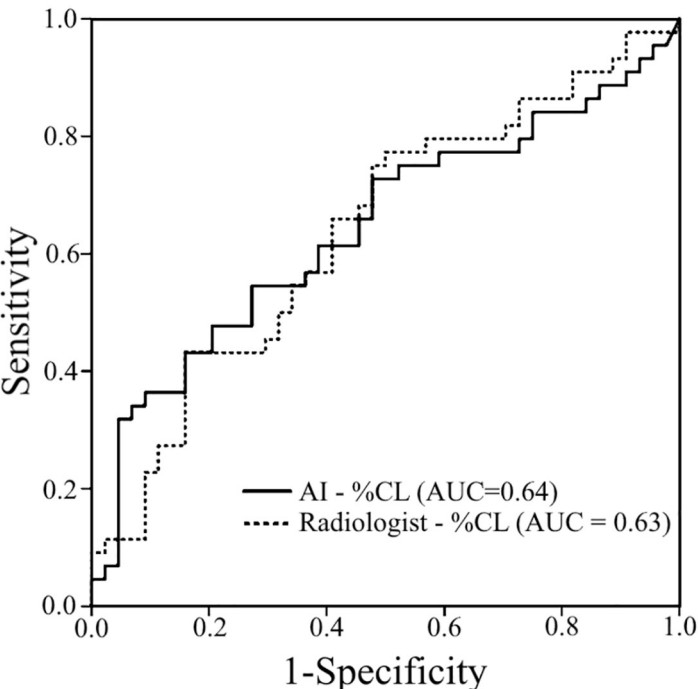

**Fig 3. Receiver operating characteristic curves of the radiologist-obtained (AUC = 0.639 [0.52–0.75]) and AI-assisted (AUC = 0.639 [0.53–0.76]) quantifications for the prediction of critical outcomes (AI = artificial intelligence; %CL = fraction of compromised lung parenchyma; AUC = area under the receiver operating characteristic curve).**

CT manifestations on CT images [9]. More recent studies have focused on the quantitative analysis of chest imaging findings to predict the disease severity or subjects' outcomes [21]. Our findings revealed a comparable or an even better performance of AI-assisted quantifications in predicting critical outcomes compared to that of radiologist-obtained measurements. Further, the final model consisting of clinical (SpO2 level, immunocompromised condition, and other comorbidities) and imaging (AI-assisted %CL) parameters showed the highest accuracy (83.9%).

There is a paucity of evidence that confirms the superiority of AI-assisted quantification over conventional semi-quantitative CT scores [22]. The semi-quantitative scoring systems are unable to distinguish <25% of lung involvement in one zone, limiting their performance in predicting outcomes [5]. Moreover, these scoring tools are used visually, which can be affected by confounding factors, including radiologist experience, and require more time. Taking all of

**Table 2. Univariate and multivariate regression analyses of the clinical and paraclinical findings for predicting critical outcomes.**

| Variable | Univariate regression | | | Multivariate regression model | | |
|---|---|---|---|---|---|---|
| | OR | 95%CI | *p*-value | OR | 95%CI | *p*-value |
| $AI - \%CL \geq 35\%$ | 4.0 | 1.5 – 11.0 | 0.007 | 17.0 | 2.2 – 128.7 | 0.006 |
| $SpO_2 < 88\%$ | 23.1 | 6.8 – 77.9 | <0.001 | 33.6 | 6.5 – 173.2 | <0.001 |
| Immunocompromised | 7.0 | 1.4 – 33.5 | 0.01 | 8.1 | 1.0 – 63.7 | 0.04 |
| Comorbidities | 2.7 | 1.1 – 6.4 | 0.02 | 15.2 | 2.3 – 98.8 | 0.004 |

OR = odds ratio; CI = confidence interval; AI = artificial intelligence; %CL = fraction of compromised lung parenchyma; SpO2 = oxygen saturation.

**Table 3. Sensitivity, specificity, PPV, NPV, and accuracy of clinical, radiology, and AI findings in predicting the critical cases.**

| Findings | Sensitivity | Specificity | PPV | NPV | Accuracy |
|---|---|---|---|---|---|
| AI − %CL | 55.6 [40.00 to 70.36] | 68.9 [53.35 to 81.83] | 64.1 [51.81 to 74.78] | 60.78 [51.43 to 69.41] | 62.2 [51.38 to 72.23] |
| Radiologist-%CL | 53.3 [37.87 to 68.34] | 64.4 [48.78 to 78.13] | 60.0 [48.16 to 70.77] | 58.0 [48.56 to 66.89] | 58.9 [48.02 to 69.16] |
| SpO2 < 88% | 69.8 [53.87 to 82.82] | 90.9 [78.33 to 97.47] | 88.4 [74.27 to 95.12] | 75.5 [65.93 to 83.03] | 80.5 [70.57 to 88.19] |
| Immunocompromised | 24.4 [12.88 to 39.54] | 95.6 [84.85 to 99.46] | 84.6 [56.36 to 95.91] | 55.8 [51.43 to 60.17] | 60.0 [49.13 to 70.19] |
| Comorbidities | 60.0 [44.33 to 74.30] | 64.4 [48.78 to 78.13] | 62.8 [51.58 to 72.78] | 61.7 [51.46 to 71.00] | 62.2 [51.38 to 72.23] |
| Clinical model(SpO2 < 88%, Immunocompromised, Comorbidities) | 79.1 [63.96 to 89.96] | 86.4 [72.65 to 94.83] | 85.0 [72.62 to 92.37] | 80.8 [70.01 to 88.42] | 82.8 [73.16 to 90.02] |
| Radiology and clinical model | 76.7 [63.96 to 89.96] | 88.9 [75.44 to 96.21] | 86.8 [74.60 to 94.03] | 80.0 [70.60 to 88.66] | 82.9 [74.48 to 90.91] |
| AI and clinical model | 79.1 [61.37 to 88.24] | 88.6 [78.33 to 97.47] | 87.2 [76.16 to 95.52] | 81.2 [69.75 to 87.40] | 83.9 [74.48 to 90.91] |

PPV: positive predictive value; NPV: negative predictive value; AI = artificial intelligence; %CL = fraction of compromised lung parenchyma; SpO2 = oxygen saturation.

this into consideration, AI-assisted models are preferred for medical applications and risk stratification.

Many previous studies have investigated the quantitative analysis of chest CT images by a radiologist. For instance, Lanza et al. showed an AUC of 0.83 for the quantitative radiologist-obtained measurement in predicting the outcomes of intubation. They also reported a median time of 11 min for segmentation [23].

Different AI models have been proposed for the quantitative analysis of images to predict clinical outcomes. Similar to our model, AI quantification and baseline clinical and laboratory data were used in a previous study to predict the prognosis of subjects with COVID-19 (2). The authors found that both CT visual score and AI-based quantification were independent predictors of the outcomes. Their final model using a combination of clinical, laboratory, and radiological parameters showed a sensitivity of 88%, a specificity of 78%, and an accuracy of 81% in identifying critical subjects, which is similar to our findings [2]. The consolidation volume percentage showed the highest AUC (0.75) in predicting critical outcomes in a previous AI-aided study; however, only age and diabetes remained significant in the multivariate analysis [22]. Two other studies also compared the performance between AI quantifications and radiologist-assessed scores in predicting adverse subject outcomes [24, 25]. Another machine learning study used a U-Net model for lung segmentation and found that the percentage of non-lesion lung volume is negatively associated with unfavorable outcomes [26].In a separate AI investigation, findings revealed that subjective severity assessment, deep learning-based features, and radiomics demonstrated predictive capabilities for subject outcome (AUC: 0.76, AUC: 0.88, AUC: 0.83, respectively) as well as the necessity for ICU admission (AUC: 0.77, AUC: 0.80, AUC: 0.82, respectively) [27]. A deep learning tool named LungQuant was employed to characterize lung parenchyma in COVID-19 pneumonia. The AUC values for percentage of lung involvement and type of lesion were reported as 0.98 and 0.85, respectively [28]. In another investigation, AI was harnessed to identify pulmonary vascular-related structures (VRS). This study revealed a correlation between the intensity of care required and an increase in VRS, which emerged as an independent explanatory factor for mortality [29].

The AI system's comparable performance against radiologist-assessed values in predicting clinical outcomes could represent a game-changer for resource-constrained settings [25]. Although the accuracy of AI is comparable to that of a radiologist, a noteworthy drawback of the radiologist's measurement approach is its time-intensive nature, requiring approximately 20 minutes for each subject, and the associated complexity. These automated models with acceptable accuracy can help medical teams with clinical judgment and treatment approaches that minimize the adverse events and accordingly maximize the healthcare system's efficiency. Further, the addition of clinical data to imaging findings significantly increased the model's predictive performance; moreover, automated models using clinical and imaging parameters can be used by non-radiologists in emergencies when a radiologist is unavailable or overburdened [9]. Another advantage of using AI in COVID-19 management lies in the fact that the score is automated and quantitative and can be obtained rapidly, enhancing its application in this pandemic.

Strengths of this study are as follows. We have comprehensively included all demographic, clinical, laboratory, and radiologic findings to propose the best predictive model. Our proposed model possesses high accuracy in distinguishing high-risk subjects. This model could be easily used by frontline physicians in COVID-19 pandemic and will help prioritizing subjects for intensive care and more aggressive treatment. Therefore, both physicians and subjects will benefit from this study. Also, while some studies train, validate, and test their models only on the same dataset, we tried to perform prognosis analysis on an unseen, third-party dataset. This strategy addresses the limited generalizability issue of the AI-assisted models and enables reporting an unbiased estimate of how good these models perform when tested on data gathered from other sources.

This study had several limitations. First, this study was conducted in a single center and had a retrospective design, which limited the generalization of our findings. Further prospective and multicentric studies on a larger population are required to validate the predictive ability of the model. In addition, the AI system used herein was unable to detect the type of radiological findings. Finally, our study lacks a radiomics analysis, which could extract more quantitative features from medical images.

## Conclusion

In conclusion, AI-assisted measurements are as robust as quantitative radiologist-obtained measurements in predicting adverse outcomes. We strongly recommend that subjects with an AI-assisted %CL of $\geq$35%, SpO2 level of <88%, immunocompromised condition, and other comorbidities be considered as high-risk subjects for further management and treatment planning.

## Author Contributions

**Conceptualization:** Arvin Arian.

**Data curation:** Mohammad-Mehdi Mehrabi Nejad, Mostafa Zoorpaikar.

**Methodology:** Navid Hasanzadeh, Saman Sotoudeh-Paima, Masoumeh Gity, Hamid Soltanian-Zadeh.

**Resources:** Mohammad-Mehdi Mehrabi Nejad, Mostafa Zoorpaikar, Shahriar Kolahi.

**Software:** Navid Hasanzadeh, Saman Sotoudeh-Paima.

**Supervision:** Hamid Soltanian-Zadeh.

**Validation:** Shahriar Kolahi, Masoumeh Gity.

**Visualization:** Navid Hasanzadeh.

**Writing – original draft:** Arvin Arian, Mostafa Zoorpaikar, Saman Sotoudeh-Paima.

**Writing – review & editing:** Hamid Soltanian-Zadeh.

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
