## [Decision Letter · Decision Letter 0]

26 Jun 2023

PONE-D-23-07250Accuracy of Artificial Intelligence-Aided CT Quantification in Predicting the Prognosis of Patients with COVID-19PLOS ONE

Dear Dr. Soltanian-Zadeh,

Thank you for submitting your manuscript to PLOS ONE. After careful consideration, we feel that it has merit but does not fully meet PLOS ONE’s publication criteria as it currently stands. Therefore, we invite you to submit a revised version of the manuscript that addresses the points raised during the review process.

We look forward to receiving your revised manuscript.

Kind regards,

Lorenzo Faggioni, M.D., Ph.D.

Academic Editor

PLOS ONE

Journal Requirements:

2. Please include a complete ethics statement in the Methods section, including the approval number and title of the approved study, and how this study fits within the larger approved study.

4. Please update your submission to use the PLOS LaTeX template. The template and more information on our requirements for LaTeX submissions can be found at http://journals.plos.org/plosone/s/latex.

Reviewers' comments:

Reviewer's Responses to Questions

**Comments to the Author**

1. Is the manuscript technically sound, and do the data support the conclusions?

Reviewer #1: Yes

2. Has the statistical analysis been performed appropriately and rigorously? 

Reviewer #1: Yes

3. Have the authors made all data underlying the findings in their manuscript fully available?

Reviewer #1: Yes

4. Is the manuscript presented in an intelligible fashion and written in standard English?

Reviewer #1: Yes

5. Review Comments to the Author

Reviewer #1: Dear authors,

Thank you for raising a such important topic, that is the differentiation of COVID-19 pneumonia from other lung disease using radiomics and machine learning. However, there are some major issues that should be addressed by the authors:

- A key result of the study is that AI-assisted measurements are similar to quantitative radiologist-obtained measurements in determining lung involvement in COVID-19 patients. As highlighted in the results, the combination of clinical and AI findings showed the highest predictive values. However, considering that also clinical and radiologist measurements achieved optimal predictive value, it may not be enough. To address this point, the authors should explore the limitations of radiological measurements compared to AI-based measurements.

Indeed, in the 2.4 Radiologist-Obtained Quantification it is reported only the adopted image analysis software. Could be interesting to report how the software provides compromised lung parenchyma and total lung segmentation (e.g. density mask technique?). Was the segmentation automated, semi-automated, manual? Was the time required for each segmentation recorded?

Other minor issues:

- Introduction: Sars Cov 2 virus was first identified in December 2019, that is significantly more than “over a year”, please rephrase.

The introduction could benefit from a more detailed summarize of the different semiquantitative and quantitative models (https://doi.org/10.5114%2Fpjr.2020.98009).

- Methods: The entire section 2.1 Study design and Population should be restructured: it should start with details about the study design (e.g. that is a retrospective study). Define inclusion and exclusion criteria and detailed information about the source of both datasets.

As mentioned above, major detail are required in the 2.4 paragraph.

- Discussion: Similarly to introduction, it should be interesting to provide more examples about limitations of semiquantitative scoring: Scapicchio et al. (http://dx.doi.org/10.1186/s41747-023-00334-z) demonstrated a wide variability in qualitative and semiquantitative scoring and provide a deep-learning based characterization of lung involvement. A similar approach was followed by Arru et al. (http://dx.doi.org/10.1016/j.clinimag.2021.06.036). A comparison with these two works or a discussion other kind of quantification (Romei et al. quantified interstitial lung disease and vessel volume in patients affected by COVID-19 http://dx.doi.org/10.1007/s00330-021-08485-6) would be of benefit for the discussion of the study results.

6. PLOS authors have the option to publish the peer review history of their article (what does this mean?). If published, this will include your full peer review and any attached files.

Reviewer #1: No

---

## [Decision Letter · Decision Letter 1]

13 Nov 2023

Accuracy of artificial intelligence CT quantification in predicting COVID-19 subjects’ prognosis

PONE-D-23-07250R1

Dear Dr. Soltanian-Zadeh,

We’re pleased to inform you that your manuscript has been judged scientifically suitable for publication and will be formally accepted for publication once it meets all outstanding technical requirements.

Kind regards,

Lorenzo Faggioni, M.D., Ph.D.

Academic Editor

PLOS ONE

Reviewers' comments:

Reviewer's Responses to Questions

**Comments to the Author**

1. If the authors have adequately addressed your comments raised in a previous round of review and you feel that this manuscript is now acceptable for publication, you may indicate that here to bypass the “Comments to the Author” section, enter your conflict of interest statement in the “Confidential to Editor” section, and submit your "Accept" recommendation.

Reviewer #1: All comments have been addressed

2. Is the manuscript technically sound, and do the data support the conclusions?

Reviewer #1: Yes

3. Has the statistical analysis been performed appropriately and rigorously? 

Reviewer #1: Yes

4. Have the authors made all data underlying the findings in their manuscript fully available?

Reviewer #1: Yes

5. Is the manuscript presented in an intelligible fashion and written in standard English?

Reviewer #1: Yes

6. Review Comments to the Author

Reviewer #1: (No Response)

7. PLOS authors have the option to publish the peer review history of their article (what does this mean?). If published, this will include your full peer review and any attached files.

Reviewer #1: No

---

## [Editor Report · Acceptance letter]

28 Nov 2023

PONE-D-23-07250R1 

Accuracy of artificial intelligence CT quantification in predicting COVID-19 subjects’ prognosis 

Dear Dr. Soltanian-Zadeh:

I'm pleased to inform you that your manuscript has been deemed suitable for publication in PLOS ONE. Congratulations! Your manuscript is now with our production department. 

Kind regards, 

on behalf of

Dr. Lorenzo Faggioni 

Academic Editor

PLOS ONE